# Occurrence, Virulence and Antimicrobial Resistance-Associated Markers in *Campylobacter* Species Isolated from Retail Fresh Milk and Water Samples in Two District Municipalities in the Eastern Cape Province, South Africa

**DOI:** 10.3390/antibiotics9070426

**Published:** 2020-07-21

**Authors:** Aboi Igwaran, Anthony Ifeanyi Okoh

**Affiliations:** 1SAMRC Microbial Water Quality Monitoring Centre, University of Fort Hare, Alice 5700, South Africa; AOkoh@ufh.ac.za; 2Applied and Environmental Microbiology Research Group (AEMREG), Department of Biochemistry and Microbiology, University of Fort Hare, Private Bag X1314, Alice 5700, South Africa

**Keywords:** campylobacteriosis, contamination, infection, resistance, virulence, waterborne

## Abstract

*Campylobacter* species are among the major bacteria implicated in human gastrointestinal infections and are majorly found in faeces of domestic animals, sewage discharges and agricultural runoff. These pathogens have been implicated in diseases outbreaks through consumption of contaminated milk and water in some parts of the globe and reports on this is very scanty in the Eastern Cape Province. Hence, this study evaluated the occurrence as well as virulence and antimicrobial-associated makers of *Campylobacter* species recovered from milk and water samples. A total of 56 water samples and 72 raw milk samples were collected and the samples were processed for enrichment in Bolton broth and incubated for 48 h in 10% CO_2_ at 42 °C under microaerobic condition. Thereafter, the enriched cultures were further processed and purified. After which, presumptive *Campylobacter* colonies were isolated and later confirmed by PCR using specific primers for the detection of the genus *Campylobacter*, target species and virulence associated genes. Antimicrobial resistance profiles of the isolates were determined by disk diffusion method against a panel of 12 antibiotics and relevant genotypic resistance genes were assessed by PCR assay. A total of 438 presumptive *Campylobacter* isolates were obtained; from which, 162 were identified as belonging to the genus *Campylobacter* of which 36.92% were obtained from water samples and 37.11% from milk samples. The 162 confirmed isolates were further delineated into four species, of which, 7.41%, 27.16% and 8.64% were identified as *C*. *fetus*, *C*. *jejuni* and *C*. *coli* respectively. Among the virulence genes screened for, the *iam* (32.88%) was most prevalent, followed by *flgR* (26.87%) gene and *cdtB* and *cadF* (5.71% each) genes. Of the 12 antibiotics tested, the highest phenotypic resistance displayed by *Campylobacter* isolates was against clindamycin (95.68%), while the lowest was observed against imipenem (21.47%). Other high phenotypic resistance displayed by the isolates were against erythromycin (95.06%), followed by ceftriaxone (93.21%), doxycycline (87.65%), azithromycin and ampicillin (87.04% each), tetracycline (83.33%), chloramphenicol (78.27%), ciprofloxacin (77.78%), levofloxacin (59.88%) and gentamicin (56.17%). Relevant resistance genes were assessed in the isolates that showed high phenotypic resistance, and the highest resistance gene harbored by the isolates was *catII* (95%) gene while *VIM*, *KPC*, *Ges*, *bla-_OXA_-*48-like, *tetC*, *tetD*, *tetK*, *IMI* and *catI* genes were not detected. The occurrence of this pathogen and the detection of virulence and antimicrobial resistance-associated genes in *Campylobacter* isolates recovered from milk/water samples position them a risk to human health.

## 1. Introduction

*Campylobacter* species are frequent enteric pathogens that cause diarrhea [1,2], and these pathogens are of great significant to public health due to the increasing number of species implicated in human infections [3]. Most campylobacteriosis cases are through consumption of contaminated food [4], unpasteurized milk [5] and contaminated water [6]. Water is important to life, but a lot of persons lack access to safe and clean water. As a result of this problem, many persons die of waterborne bacterial infections [7]. Waterborne infection is a worldwide burden that is approximated to cause millions of deaths annually and daily cases of illness including systematic illnesses, diarrhea and gastroenteritis [8,9]. Water sources, including rivers, lakes, streams and ponds, have numerous potential contamination sources such as faecal droppings of animals on pasture, direct faecal contamination by wild birds within the watersheds [10] and discharge of poorly treated wastewater effluents or non-disinfected sewage [11]. In South Africa, gastroenteritis, viral hepatitis, cholera, typhoid fever and dysentery are among waterborne infections that pose a high risk to the citizens [12]. Gastroenteritis is one of the major symptoms of campylobacteriosis. Consumption of unpasteurized milk from cows have been reported to be implicated in human campylobacteriosis [13,14]. Globally, milk consumption is predicted to be in billions of liters and most of which are consumed as pasteurized. Though, in recent years, there has been a rise in the rate of consumption of unpasteurized milk compared to pasteurized milk [15]. Raw milk consumption is highly unsafe for infants, pregnant women, the aged and immunocompromised persons [16]; and the propensity towards consumption of unpasteurized raw milk is due to its health benefits, taste and higher nutritional qualities.

Thus, consumption of unpasteurized raw milk positions the consumers at high risk of ill-health which could leads to diseases outbreaks. Raw milk is sometimes contaminated with pathogenic microbes and this usually occurs from sick animals or from environmental sources [14]. Several reports, including the studies of Artursson et al. [17] and Del Collo et al. [18], have also detected *Campylobacter* species in raw milk samples. *Campylobacter* species are among the several pathogens that sometimes contaminate raw milk [19,20]. In several parts of the world, *Campylobacter* species have been reported to be implicated in disease outbreaks and *Campylobacter* infections are of varying severity ranges from abdominal pains, vomiting, nausea, fever and diarrhea [21,22,23]. In extreme cases, acute phase of *Campylobacter* infection is followed by sequelae: Guillain-Barré syndrome or even death [24]. Some *Campylobacter* species reported to be implicated in human infections includes *C. fetus*, *C. jejuni*, *C. coli* and *C. lari* [25]. In addition to the burden of infections caused by these bacteria pathogens, the spread of antibiotic resistant-*Campylobacter* strains is another burden of public health plight which might be more severe in developing countries where there is largely uncontrolled use of antimicrobials [26,27]. Antibiotic resistance is known as a One Health concern due to the rapid emergence and spreading of resistant bacteria and resistant genes on a global scale [28]. Antibiotic resistant bacteria (ARB) are majorly disseminated through discharge of animal manure, human waste and wastewater effluents into the environment which can lead to the development of antibiotic-resistant genes (ARGs) in the exposed bacteria [29]. Antibiotic resistance can be mediated through vertical gene transfer or through genetic exchanges between and within bacteria species [30]. ARGs are emerging environmental pollutants and aquatic environments are known as one of the major reservoirs of ARB and ARGs [31]. Hence, this study evaluated the occurrence as well as virulence and antimicrobial resistance-associated markers of *Campylobacter* species recovered from milk and water samples in the Chris Hani and Amathole District Municipalities in the Eastern Cape Province, South Africa.

## 2. Material and Methods

### 2.1. Ethical Clearance

Ethical clearance was applied for the study and granted by the University of Fort Hare research ethics committee with certificate reference number: OKO021IGW01.

### 2.2. Description of Study Area

The study was carried out in Chris Hani and Amathole District Municipalities, in the Eastern Cape Province, South Africa with geographical co-ordinates “31.8743° S, 26.7968° E′′ and “32.5842° S, 27.3616° E′′ respectively.

### 2.3. Collection of Samples

A total of 128 samples were collected, comprising of 40 water samples from rivers and 16 water samples from pond/dams (used for irrigation), 40 milk samples from cow/bulk milk tanks from farms, 15 milk samples from cars/roadside, 9 milk samples from retail markets and 8 milk samples from butcheries. The water samples were collected in sterile 1L polypropylene bottles while the milk samples were collected in sterile 250 mL polypropylene bottles. All the samples were collected in Amathole and Chris Hani District Municipalities in the Eastern Cape Province, South Africa, transported in cooler box with ice and were analysed within 6 h of collection.

### 2.4. Isolation of Campylobacter Species from Water Samples

The method described by Van Dyke et al. [32] was adopted for *Campylobacter* isolation. Briefly, 1000 mL of water samples were filtered through nitrocellulose membrane filters (0.45-μm pore size). The filter papers were picked with sterilized forceps, added into 20 mL of Bolton selective enrichment broth supplemented with Bolton broth selective supplement with 5% (*v*/*v*) defibrinated horse blood and incubated in 10% CO_2_ at 42 °C for 48 h under microaerophilic condition in HF151UV CO_2_ incubator. Thereafter, a loopful from the enriched cultures were streaked onto modified cefoperazone deoxycholate agar (mCCDA) plates supplemented with antibiotic selective supplement (CCDA selective supplement (cefoperazone and amphotericin)), incubated as before. Presumptive *Campylobacter* colonies were picked and re-streaked onto blood agar plates supplemented with 7% (*v*/*v*) defibrinated horse blood and incubated as before.

### 2.5. Isolation of Campylobacter Species from Milk Samples

The milk samples were processed following the method previously described by Bianchini et al. [33]. Briefly, 20 mL of the milk samples was introduced into 200 mL of Bolton selective enrichment broths (1:10 ratio) to which Bolton antibiotic supplement with 5% (*v*/*v*) defibrinated horse blood were added and incubated at 42 °C for 48 h under microaerophilic atmosphere in 10%CO_2_ in HF151UV CO_2_ incubator. Thereafter, isolation and purification process described in Section 2.4 were followed.

### 2.6. DNA Extraction

Bacteria DNA was extracted by boiling method following the method of Sierra-Arguello et al. [34] with slight modification. Briefly, single *Campylobacter* colonies from the blood agar plates were isolated and grown in 5 mL of Tryptone Soya Broth (TSB) incubated for 48 h at 42 °C in 10%CO_2_ in a HF151UV CO_2_ incubator. From which, 1 mL of the broth was centrifuged for 5 min at 12,800 rpm and the supernatants were decanted and the cells were suspended in 400 µL of sterile distilled water in 1.5 mL Eppendorf tubes. The suspensions were boiled at 100 °C for 10 min in a heating block and the cell debris were removed by centrifugation for 5 min at 12,800 rpm and the supernatants were collected and stored at −20 °C until ready for use.

### 2.7. Molecular Confirmation Characterization and Amplification of Virulence Genes

Presumptive *Campylobacter* isolates were confirmed by PCR for identification of the genus *Campylobacter* targeting a 439 base pairs of part of 16S rRNA gene as reported by Moreno et al. [35]. The confirmed *Campylobacter* isolates were further delineation into *C. jejuni*, *C*. *lari*, *C. fetus* and *C*. *coli* using the primer sets as listed in Appendix A targeting *cj0414*, *glyA*, *cstA* and *asK* genes respectively [36] and primers specific for virulence markers responsible for invasion (*iam*) gene [37], invasion protein gene (*ciaB*) [38] colonization gene (*flaA*), adherence (*cadF*) gene and toxin production (*cdtB*) gene [3] and flagella synthesis and modification (*flgR*) gene [39] by PCR. Both multiplex and singleplex PCR were carried out in a 25 μL reaction volume (1.0 μL of each PCR primer, 12.5 µL master mix (Inqaba Biotec, South Africa), 5.0 μL of extracted DNA and 5.50 μL of nuclease free water). The amplified PCR products were visualized by gel electrophoresis in a 1.5% (*w*/*v*) agarose stained with ethidium bromide in 5xTAE buffer. 

### 2.8. Antibiotic Resistance of Campylobacter Isolates

The disc diffusion technique on Mueller Hinton agar plates supplemented with 5% defibrinated horse blood was used to characterized the sensitivity of *Campylobacter* isolates against antimicrobial agents [40]. In summary, bacterial growth in TSB incubated at 42 °C for 48 h in 10% CO_2_ were adjusted to 0.5 McFarland turbidity standard in sterile normal saline followed by a gentle spread of the solution with a cotton swab on the entire surface of Mueller Hinton agar plates. Afterward, antibiotic discs were impregnated on the plates and incubated for 24 h at 42 °C in CO_2_ incubator under microaerobic conditions. The selected antimicrobials used were doxycycline (30 μg), tetracycline (30 μg), ampicillin (10 μg), azithromycin (15 µg), erythromycin (15 μg), gentamicin (10 μg), clindamycin (2 μg), chloramphenicol (30 μg), ciprofloxacin (5 μg), levofloxacin (5 µg), ceftriaxone (30 μg) and imipenem (10 µg). The inhibition zones for tetracycline, doxycycline, ciprofloxacin and erythromycin were interpreted according to CLSI [40] guidelines for *Campylobacter*. As there are no guidelines available for *Campylobacter* against ampicillin, azithromycin, gentamicin, clindamycin, chloramphenicol, levofloxacin, ceftriaxone and imipenem; CLSI, [40] guideline for *Enterobacteriaceae* were used for the interpretation of results.

### 2.9. Multiple Antibiotic Resistance MAR Index

The MAR index of each of the *Campylobacter* isolates were calculated using the formula MAR = a/b as reported by Krumperman, [41]. Where a= is the number of antibiotics to which the test isolate showed resistance to and b= is the total number of antibiotics to which the test isolate has been evaluated for susceptibility.

### 2.10. Molecular Screening Of Antimicrobial Resistance Genes

The isolates that showed phenotypic resistance to the test antibiotics were subjected to molecular screening for the detection of genotypic resistance genes employing PCR method. The primer sets reported by Ng et al. [42] was used for the detection of *tetA*, *tetB*, *tetC*, and *tetD* genes, for *tetK* and *tetM* genes [43], *gyrA* gene [44], *ermB* gene [45], *catI* and *catII* genes [46] and the *aac(3)-IIa-(aacC2)* [47] and *VIM*, *KPC*, *Ges*, *bla-_OXA_-*48-like and *IMI* genes [48] and the primer sets are shown in Appendix A.

### 2.11. Statistical Analysis

Statistical analysis was carried out by Microsoft office tools.

## 3. Results

### 3.1. Molecular Identification of the Genus Campylobacter

A total of 438 presumptive *Campylobacter* isolates were obtained, from which 162 (36.99%) were identified as belonging to the genus *Campylobacter* of which 103 (36.92%) isolates out of the 279 presumptive isolates were detected in water samples from rivers/dams, and 33 (58.93%) water samples out of 56 water samples were positive for *Campylobacter*. In the milk samples, 59 (37.11%) isolates out of 159 presumptive isolates were detected to be *Campylobacter* and 19 (26.38%) milk samples out of 72 milk samples obtained from butcheries, farms, retail markets and car/roads were positive for *Campylobacter* (Figure 1). However, not all milk and water samples were positive for *Campylobacter*. Figure 1 is a pictorial representation of presumptive/confirmed *Campylobacter* isolates recovered from milk samples from different sources while Figure 2 is a representative gel picture of some PCR confirmed genus *Campylobacter*.

### 3.2. Molecular Detection of C. coli C. jejuni and C. fetus

The 162 confirmed isolates identified as belonging to the genus *Campylobacter* were further delineated into *C*. *coli*, *C*. *fetus* and *C*. *jejuni* while *C*. *lari* was not detected. The detailed distribution patterns of occurrence of the identified species is as shown in Table 1 while Figure 3, Figure 4 and Figure 5 are representative gel images of some identified *C*. *coli*, *C*. *fetus*, and *C*. *jejuni* isolates.

### 3.3. Molecular Detection of Virulence Genes in the Identified Campylobacter Species

Assessment of virulence genes were determined by PCR techniques and the virulence genes associated with toxin production (*cdtB*), invasion (*iam* and *ciaB*), adherence (*cadF* and *flaA*), and flagellia synthesis and regulator (*flgR*) genes were detected. From the six virulence genes screened for among the 70 isolates identified as *C*. *coli*, *C*. *jejuni* and *C*. *fetus* (Table 1), the *iam* (32.86%) gene was most prevalent in all the *Campylobacter* species, followed by *flgR* (20%) gene, and *cdtB* and *cadF* (5.71%) genes. The observed percentage occurrence of virulence-associated genes detected among *C*. *fetus*, C. *jejuni* and *C*. *coli* were different, except for the *ciaB* gene that was not detected in all the isolates. From the PCR results obtained, high occurrence of *iam* (35%) gene was detected in *C*. *jejuni* isolates while low incidence of *flgR* (4.55%) gene was detected in *C*. *jejuni* isolates. It was also observed that virulence-associated genes were more often detected in *C*. *coli* than in *C*. *jejuni* and *C*. *fetus*. In terms of classes of virulence genes co-harbored in the identified species, 4 (5.71%) *C. coli* isolates co-harbored the *iam* and *flaR* genes, 1 (1.43%) *C. jejuni* isolate co-harbored *iam* and *flaR* genes, 1(1.43%) *C. coli* isolates co-harbored *iam* and *cadF* genes, 1 (1.43%) *C. coli* strain co-harbored *iam*, *cadF* and *cdtB* genes and 2 (2.88%) isolates identified as *C*. *jejuni* and *C. coli* co-harbored *iam* and *cdtB* genes. The detailed distribution pattern of the virulence genes detected in *Campylobacter* species recovered from both water and milk samples are as shown in Table 2 while Figure 6 and Figure 7 are representative gel images of the detected virulence-associated genes.

### 3.4. Antibiotic Phenotypic Resistance Profiles of Campylobacter Isolates

The 162 *Campylobacter* isolates obtained from water and milk samples were tested against 12 antimicrobials agents. Of the 12 antibiotics tested, the highest phenotypic resistance displayed by *Campylobacter* isolates recovered from milk and water samples was against clindamycin (95.68%), while the lowest was observed against imipenem (21.47%). Other high phenotypic resistance displayed by tThe e isolates were against erythromycin (95.06%), followed by ceftriaxone (93.21%), doxycycline (87.65%), azithromycin and ampicillin (87.04% each), tetracycline (83.33%), chloramphenicol (78.27%), ciprofloxacin (77.78%), levofloxacin (59.88%) and gentamicin (56.17%) (Figure 8). Most of the isolates were resistance to more than three classes of antimicrobial agents and were classified as multi-drug resistance (MDR). The lowest phenotypic MDR rate observed in *C*. *coli* isolate was to CRO-E-CD-AP, in *C*. *fetus* was to CRO-E-CD-T-DXT-AP and in *C*. *jejuni* was to E-ATH-mCD-T-DXT-AP (Table 3). Mand majority of the isolates showed resistance to more than 2 to 9 classes of antimicrobial agents, and the highest resistance profiles observed in *C*. *jejuni*, *C*. *coli* and *C*. *fetus* isolates were to LEV-CRO-C-CIP-E-ATH-CD-T-GM-DXT-AP (22.86%) and LEV-CRO-C-CIP-E-ATH-IMI-CD-T-GM-DXT-AP (10%). The detailed multiple resistance patterns exhibited by *C*. *coli*, *C*. *jejuni* and *C*. *fetus* are showed in Table 3.

### 3.5. Molecular Detection of Genotypic Resistance Genes in Campylobacter Isolates

Genotypic resistance genes in *Campylobacter* isolates were detected by PCR and the prevalence of *catII* gene in chloramphenicol resistance *Campylobacter* isolates was the highest resistance gene detected, where 38 (95%) isolates identified as *C. coli*, *C. jejuni* and *C*. *fetus* harbored the *catII* gene. Tetracycline resistance genes were widespread in *C. coli*, *C. jejuni* and *C*. *fetus* where *tetA, tetB* and *tetM* were detected in 88.71%, 27.42% and 32.26% respectively. Other ARGs detected in *C. coli*, *C. jejuni* and *C*. *fetus*, including *ermB* (erythromycin resistance gene), *gyrA* (gentamycin resistance gene), *ampC* (ampicillin resistance gene) and *aac(3)-IIa-(aacC2)^a^* (gentamycin resistance gene), were 15.38%, 39.13%, 81.54% and 84.85% respectively. All *C. coli*, *C. jejuni* and *C*. *fetus* recovered from both milk and water samples were negative for *VIM*, *KPC*, *Ges*, *bla-_OXA_-*48-like, *tetC*, *tetD*, *tetK*, *IMI*, and *catI* genes. From the PCR results obtained, most of the isolates were observed to harbor multiple resistance genes and the highest number of resistance genes detected in *C*. *jejuni* isolates were *tetA*, *tetM, ampC*, *catII*, *gyrA*, *aac(3)-IIa-(aacC2)^a^* genes, in *C*. *coli* isolate were *tetA*, *tetM, ampC*, *catII*, *ermB*, *aac(3)-IIa-(aacC2)^a^* genes while in *C*. *fetus* isolates were *tetA*, *tetM, ampC*, *catII*, *ermB*, a*aac(3)-IIa-(aacC2)^a^* genes (Table 4). Detection of multiple resistance genes in the isolates indicates that the isolates simultaneously carry two or more classes of antimicrobial resistance genes. Table 4 showed the detailed pattern of multiple antibiotic resistance genes detected in *C*. *fetus*, *C. jejuni* and *C*. *coli* recovered from water and milk samples while Figure 9 and Figure 10 are representative gel electrophoreses images of the amplified PCR products.

## 4. Discussion

Globally, there is an increasing rate in the detection of *Campylobacter* species including reports from Africa, America, Asia, and Europe [49,50,51] and this is of great concern to public health [52]. *Campylobacter* species are implicated in both waterborne/milkborne infections, and it is vital to provide more information to existing reports on the risk of consumption of unchlorinated water and unpasteurized milk. Hence, this study evaluated the occurrence as well as virulence and antimicrobial resistance-associated makers of *Campylobacter* species isolated from retailed milk and water samples. Occurrence of *Campylobacter* species in water/milk samples was determined by culture-based and PCR techniques and reports on this is very scanty in the Eastern Cape Province which has the largest livestock in South Africa. In this study, *Campylobacter* was detected in 103 (36.92%) isolates recovered from water samples and 33 (58.93%) water samples out of 56 water samples were positive for *Campylobacter*. In the milk samples, 59 (37.11%) isolates were detected to be *Campylobacter* and 19 (26.38%) milk samples out of 72 milk samples were positive for *Campylobacter*. Results from this study showed that water samples were more contaminated with *Campylobacter* than the milk samples and our finding is in agreement with the report of Elmal and Can, [53]. Other studies carried out by Khan et al. [54], Szczepanska, et al. [55] and Van Dyke et al. [32] reported high detection rates of *Campylobacter* species in river water samples and our finding correspond with their reports.

Other studies conducted by Artursson et al. [17], Bianchini et al. [33] and Wysok et al. [56] have also detected *Campylobacter* species in raw milk samples and this current finding is also in line with their reports. Consumption of raw milk has been implicated in campylobacteriosis cases; a behaviour that has attracted attention lately. Worldwide, campylobacteriosis add ominously to the burden of human enteric illness [57]. In the United States and Europe, consumption of raw cow’s milk has been reported to be implicated in campylobacteriosis outbreaks [58,59]. In the Limpopo Province, South Africa, *Campylobacter* species were reported to be common causes of gastroenteritis in children [60,61] and the detection of these pathogens in retail raw milk/water samples in this study area, position them as a public health concern of provincial interest. In the water samples, occurrence of *C*. *jejuni* was most prevalent with percentage detection rate of 38.83%, followed by 7.77% for *C. coli* and 5.83% for *C. fetus* and a similar result was reported in the studies of Denis et al. [62], Pérez-Boto et al. [63] and Szczepanska et al. [55]. In the milk samples, *C*. *coli* and *C*. *fetus* were detected to be most prevalent and our finding is similar with the report of Mabote et al. [64] but contrary to the reports of Andrzejewska et al. [65], Kabir et al. [66] and Rahimi et al. [67].

The detection of pathogenic *Campylobacter* species in river water samples and retailed milk samples highlights the significance of river and raw milk as a potential reservoir of *Campylobacter* species. Of the 70 isolates identified as *C*. *coli*, *C*. *fetus* and *C*. *lari*, the major virulence genes detected were the *iam*, *flgR*, *cdtB* and *cadF* genes (Table 2). Our study showed the distribution patterns of the *iam* gene among the *Campylobacter* species and the *iam* gene is a virulence marker responsible for invasion of host cell. The *iam* gene was detected in both *C*. *jejuni* and *C*. *coli* isolates recovered from water and milk samples and this result is akin with the reports of Ghorbanalizadgan et al. [68], Pandey et al. [69] and Wysok et al. [70]. In another study of Bardoň et al. [71], the *iam* gene was majorly detected in *C*. *jejuni* than in *C*. *coli* and our result is contrary to this report. The *cdtB* gene is another virulence gene assessed responsible for toxin production, and studies have detected the *cdtB* gene in *C*. *jejuni* and *C*. *coli* strain recovered from beef, raw milk and pork [65], from chicken [60], from humans [72] and from cows’ cervical mucus [73] and our finding is also in line with these reports. Another virulence gene assessed was the *flgR* gene, the *flgR* gene was found in 50% of *C*. *coli*, 41.67% of *C*. *fetus* and 5% of *C*. *jejuni* and the *flgR* gene were detected in 5.56% water isolates and 62.5% in milk isolates (Table 2) and there is large variability of detection of *flgR* gene between water and milk samples. The study of Modi et al. [3] has also detected the *flgR* gene in *C*. *coli* and our finding correspond with this report. In this study area, no study has reported the detection of *flgR* gene in *Campylobacter* species. Furthermore, there are few reports on the detection of *flgR* gene in *Campylobacter* isolates recovered from milk and water. The *flgR* gene is liable for phase variation—a mechanism that help the bacteria to modify the antigenic make-up of its surface to adapt to new hosts [74]. Another gene detected was the *cadF* gene, and the *cadF* gene is a virulence gene that helps in binding to the intestinal epithelial cells [75,76]. The *cadF* gene (5.71%) was detected in *C*. *coli* and *C*. *fetus* recovered from water and milk samples. In the studies of Lluque et al. [77], Wieczorek et al. [78] and Selwet et al. [79], the *cadF* gene was detected in *Campylobacter* species from a Peruvian pediatric cohort, from meat samples and from *Campylobacter* isolates isolated from dogs and these reports corroborate our finding. The presence of one or more virulence genes in the *Campylobacter* genome give rise to the incidence of human infection (Abu-Madi et al. [80]). In our study, some *Campylobacter* species were observed to harbor multiple virulence genes and several studies including the studies of Aslantaş [81], Redondo et al. [82], Samad et al. [83] and Wei et al. [84] have also detected multiple virulence genes in *Campylobacter* species and our finding also corroborate with their reports. Detection of these virulence genes in *Campylobacter* isolates recovered from retail milk and water samples position them a risk to human health and continuous consumption of raw milk in the study area may put people at high risk of ill-health. The confirmed isolates were tested against a panel of 12 antibiotics, and the highest phenotypic resistant displayed by the *Campylobacter* species was to clindamycin (95.68%) and the lowest was observed against imipenem (21.47%) (Figure 8). Multidrug resistance to azithromycin, ampicillin and ciprofloxacin were observed in the study of Martín-Maldonado et al. [85] and in this study we also observed similar multiple resistance pattern (Table 3). In this study, high phenotypic *Campylobacter* resistance to ciprofloxacin (77.78%) was observed and this finding corroborate with the report of Meistere et al. [86], who also reported high phenotypic resistance to ciprofloxacin (93.6%) in *Campylobacter* isolates.

High resistance rate to tetracycline (83.33%) were also observed in the isolates and this result is similar with the report of Elhadidy et al. [87] who also reported high phenotypic *Campylobacter* resistance rate of 81.4% to tetracycline. Furthermore, high susceptibility level was observed against imipenem and our result also corroborate with this report of Noreen et al. [88]. Analysis of the MAR indices of the *Campylobacter* isolates showed that MAR indices values were all greater than 0.2 (Table 3). A MAR index value greater than 0.2 is said to have originated from commercial swine, poultry farms, dairy cattle and humans where antibiotics are often used and are at high-risk sources of antibiotic contamination [89,90]. The highest MAR value indices values were to twelve of the antimicrobials tested (LEV-CRO-C-CIP-E-ATH-IMI-CD-T-GM-DXT-AP). In this study, high resistance rates were observed against erythromycin (95.06%), ampicillin (87.04% each), tetracycline (83.33%), chloramphenicol (78.27%), ciprofloxacin (77.78%) and gentamicin (56.17%). Our result corresponds with the report of Abbasi et al. [91] who also observed high phenotypic *Campylobacter* isolates resistant to tetracycline, ciprofloxacin and erythromycin. The report of Nizar et al. [92] also showed high *Campylobacter* resistant to gentamycin (25.6%) and our finding is also in line with their report. Another study of Premarathne et al. [93] also observed high *Campylobacter* resistant to ampicillin and this report correspond with our result. Genotypic antimicrobial resistance genes were also determined by PCR and high *gyrA* gene (39.13%) was detected in *Campylobacter* isolates and our finding corroborate with the report of Meistere et al. [86], in which high *gyrA* gene was detected in their study. Primarily, macrolides remain the frontline antibiotic use for the treating of campylobacteriosis. However, in many countries, there have been reports on progressive increase in *Campylobacter* resistance to macrolide and this is a growing health threat concern of global concern [94]. In our study, erythromycin resistance gene *erm* (*B*) was detected in *C*. *coli* and *C*. *jejuni* isolates and our finding corroborate with the report of Liu et al. [95]. The *tet*-genes are other genes assessed and *tetA* gene is among the *tet-*genes responsible for tetracycline resistance and in our study, high rate of *tetA* (88.71%) gene was detected in tetracycline resistant-*Campylobacter* isolates and a similar result was also reported in the study of Divsalar et al. [96]. Furthermore, the high rate of *ampC* gene was also detected in ampicillin resistant-*Campylobacter* isolates and the detection of multiple resistance genes in *Campylobacter* isolates might limit the treatment option for campylobacteriosis cases.

## 5. Conclusions

The key step in the prevention of *Campylobacter* infection is monitoring of this pathogen that pose a great menace to human health. Our finding reveals that *Campylobacter* strains with important pathogenic factors responsible for toxin production (*cdtB*), invasiveness (*iam*, *ciaB*), motility (*flaA*, *flgR*) and adherence (*cadF*) were detected in the *Campylobacter* isolates recovered from river and milk samples. This study also highlights the importance of monitory of the spread of antibiotic resistant-*Campylobacter* isolates recovered from water and retail milk samples which will help determined the risk poses to human if appropriate measure is not put to hurt the distribution patterns. Furthermore, high rates of multiple phenotypic and associated genotypic antibiotic resistance genes were detected and this might further limit treatment options for *Campylobacter* infections.

## Figures and Tables

**Figure 1 antibiotics-09-00426-f001:**
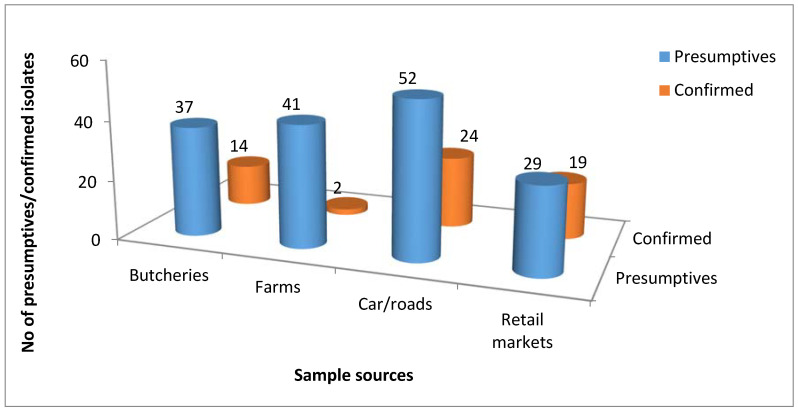
A pictorial representation of presumptive/confirmed *Campylobacter* isolates recovered from milk samples from different sources.

**Figure 2 antibiotics-09-00426-f002:**
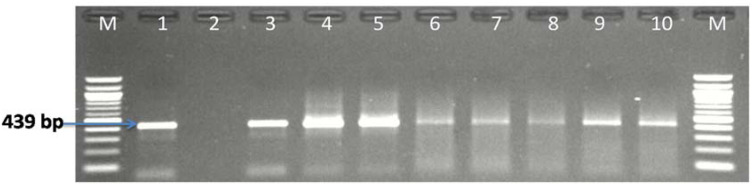
A representative gel image of PCR confirmed genus *Campylobacter*. Lane M: (100 bp DNA ladder), lane 1: positive control (*C. jejuni* ATCC 3356), lane 2: negative control, lane 3–10: some positive *Campylobacter* isolates.

**Figure 3 antibiotics-09-00426-f003:**
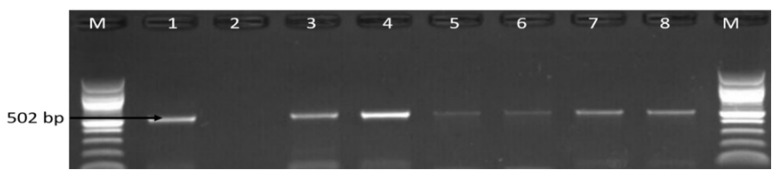
Gel image of PCR detected *aspK* (502 bp) gene of *C*. *coli*. Lane M: DNA ladder (100 bp), lane 1: positive control (*C. coli* ATCC 33559), lane 2: negative control, lane 3–8: some positive *C*. *coli* isolates (502 bp).

**Figure 4 antibiotics-09-00426-f004:**
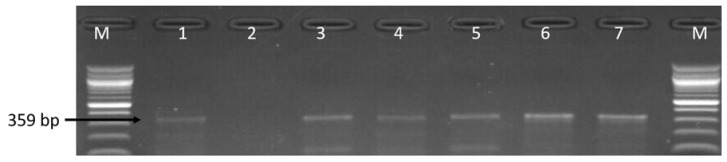
Gel electrophoresis image of PCR detected *cstA* (359 bp) gene of *C. fetus*. Lane M: molecular marker (100 bp), lane 1: positive control (*C. fetus* ATCC 27374), lane 2: negative control, lane 3–7: some positive *C. fetus* isolates (359 bp).

**Figure 5 antibiotics-09-00426-f005:**
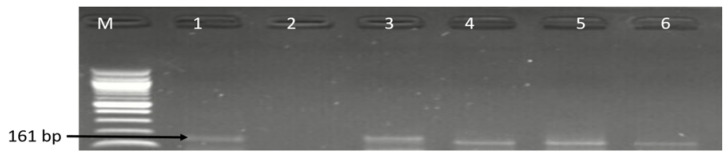
Gel electrophoresis image of identified *C*. *jejuni cj0414* gene at 161 bp. Lane M: DNA ladder (100 bp), lane 1: positive control (*C. jejuni* ATCC 33560), lane 3–6: some positive *C*. *jejuni* isolates.

**Figure 6 antibiotics-09-00426-f006:**
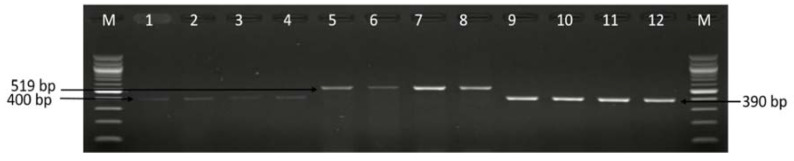
A representative gel image of some of the PCR detected *iam*, *cadF* and *fldR* genes. Lane 1–4: positive *Campylobacter* isolates that harbor *cadF* gene (400 bp), lane 5–8: positive *Campylobacter* isolates that harbor *iam* gene (519 bp), lane 9–12: positive *Campylobacter* isolates that harbored *flgR* gene (390 bp).

**Figure 7 antibiotics-09-00426-f007:**
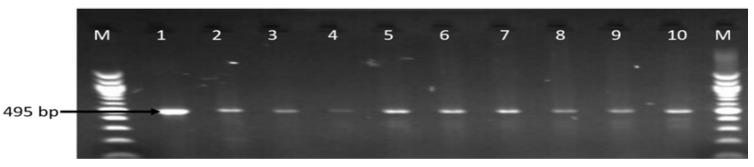
Gel image of some of the PCR detected *cdtB* gene. Lane M: DNA ladder, lane: 1–10: some positive *Campylobacter* isolates that harbor *cdtB* gene (495 bp).

**Figure 8 antibiotics-09-00426-f008:**
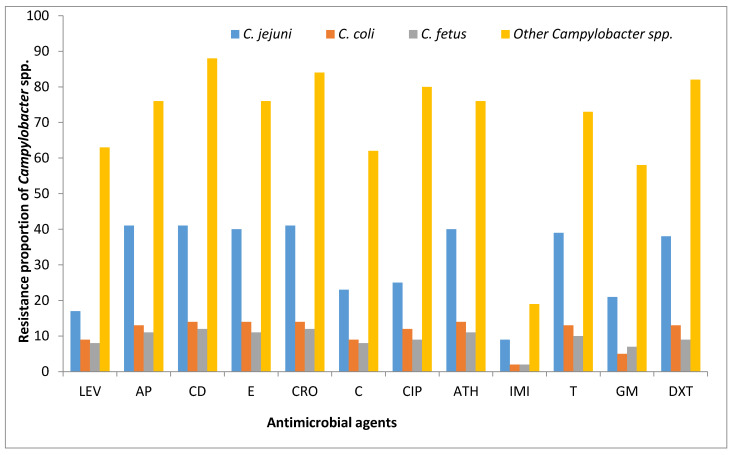
Resistance proportions of *C*. *jejuni*, *C*. *coli*, *C*. *fetus* and other *Campylobacter* species isolated from milk and water samples to 12 antimicrobial agents. Levofloxacin (LEV), ciprofloxacin (CIP), azithromycin (ATH), imipenem (IMI), ampicillin (AP), clindamycin (CD), tetracycline (TET), ceftriaxone (CRO), chloramphenicol (C), erythromycin (E), gentamicin (GM) and doxycycline (DXT).

**Figure 9 antibiotics-09-00426-f009:**
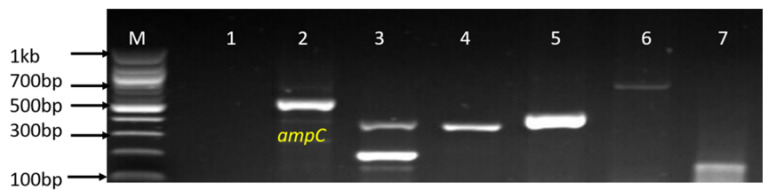
A representative electrophoresis picture of various amplified antibiotics resistance genes of *Campylobacter* isolates. Lanes M: DNA ladder (100 bp), lane 1: negative control, lane 2: *ampC* gene (530 bp), lane 3: *tetA* (201 bp) and *tetB* gene (359 bp), lane 4: *ermB* gene (320 bp), lane 5: *gyrA* gene (441 bp), lane 6: *aac(3)-IIa (aacC2)^a^* gene (740 bp) and lane 7: *tetM* gene (159 bp).

**Figure 10 antibiotics-09-00426-f010:**
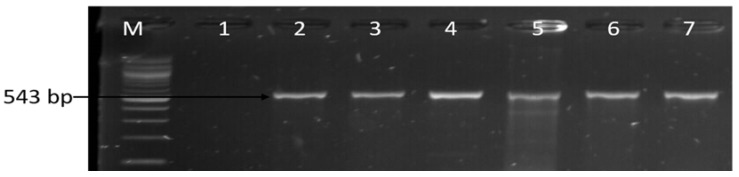
Electrophoresis gel image of PCR confirmed *catII* gene. Lane M: DNA ladder (100 bp), lane 1: negative control, lane 2–7, *Campylobacter* isolates that harbored the *catII* gene (543 bp).

**Table 1 antibiotics-09-00426-t001:** Distribution patterns of *Campylobacter* species identified in the sample sources.

Sample Sources	*C. fetus* (%)	*C. jejuni* (%)	*C. coli* (%)	*C. lari* (%)	No of Isolates That Belong to Other *Campylobacter* Species (%)
Milk	6 (10.17)	4 (6.78)	6 (10.17)	0	43 (72.88)
Water	6 (5.83)	40 (38.83)	8 (7.77)	0	49 (47.57)

**Table 2 antibiotics-09-00426-t002:** Prevalence of virulence genes detected in *Campylobacter* species.

Water Samples								Milk Samples						
*Campylobacter* spp.	No of Isolate	Virulence Genes Screened (%)						No of Isolates	Virulence Genes Screened (%)					
		*iam*	*flaA*	*cadF*	*flgR*	*cdtB*	*ciaB*		*iam*	*flaA*	*cadF*	*flgR*	*cdtB*	*ciaB*
*C. coli*	8	4 (50)	-	3 (37.)	1 (12.5)	1 (12.5)	-	6	4 (66.7)	-	-	6 (100)	-	-
*C. jejuni*	40	14 (35)	-	-	2 (5)	3 (7.5)	-	4	-	-	-	-	-	-
*C. fetus*	6	1 (16.7)	-	-	1 (16.7)	-	-	6	-	-	1 (16.7)	4 (66.7)	-	-

**Table 3 antibiotics-09-00426-t003:** Antibiotics phenotypic resistance patterns of *Campylobacter* isolates.

No	Antimicrobial Resistance Patterns	Sample Source		No of Isolates			Total	MAR Index
		Water	Milk	*C. coli*	*C. jejuni*	*C. fetus*		
1	CRO-E-CD-AP		1	1	-	-	1	0.33
2	CRO-E-CD-T-DXT-AP		1	-	-	1	1	0.5
3	E-ATH-CD-T-DXT-AP	2		-	1	-	2	0.5
4	CRO-C-E-ATH-CD-AP	1		-	1	-	1	0.5
5	LEV-C-CIP-E-ATH-CD	1		-	-	1	1	0.5
6	LEV-CRO-CIP-E-ATH-CD-AP	1		-	1	-	1	0.58
7	CRO-E-ATH-CD-T-DXT-AP	3		-	3	-	3	0.58
8	E-ATH-CD-T-GM-DXT-AP	1		-	1	-	1	0.58
9	CRO-E-ATH-CD-T-GM-AP	1		-	1	-	1	0.58
10	CRO-E-ATH-CD-T-DXT-AP	1		-	1	-	1	0.58
11	CRO-E-ATH-CD-T-GM-DXT-AP	3	2	1	3	1	5	0.67
12	CRO-C-E-ATH-CD-T-GM-AP		1	-	-	1	1	0.67
13	CRO-C-CIP-E-CD-T-DXT-AP		3	-	1	-	1	0.67
14	CRO-C-E-ATH-CD-T-DXT-AP	3		-	3	-	3	0.67
15	CRO-E-ATH-IMI-CD-T-DXT-AP	2		-	2	-	2	0.67
16	CRO-CIP-E-ATH-CD-T-DXT-AP	1		-	1	-	1	0.67
17	LEV-CRO-C-CIP-E-ATH-CD-DXT		1	1	-	-	1	0.67
18	CRO-E-ATH-IMI-CD-T-GM-AP	2		-	2	-	2	0.67
19	CRO-CIP-E-ATH-CD-T-DXT-AP	1		-	1	-	1	0.67
20	CRO-C-E-ATH-IMI-CD-T-DXT-AP		1	-	-	1	1	0.75
21	CRO-C-E-ATH-CD-T-GM-DXT-AP			-	2	-	2	0.75
22	LEV-C-CIP-E-ATH-CD-T-GM-AP		1	-	-	1	1	0.75
23	CRO-CIP-E-ATH-CD-T-GM-DXT-AP	2		1	1	-	2	0.75
24	C-CIP-E-ATH-IMI-CD-T-DXT-AP		1	-	1	-	1	0.75
25	LEV-CRO-CIP-E-ATH-CD-T-DXT-AP	1		-	1	-	1	0.75
26	LEV-CRO-C-CIP-E-ATH-CD-T-DXT-AP	1		-	1	2	3	0.83
27	LEV-CRO-CIP-E-ATH-CD-T-GM-DXT-AP	4		1	1	2	4	0.83
28	CRO-C-CIP-E-ATH-CD-T-GM-DXT-AP	1		1	-	-	1	0.83
29	CRO-CIP-E-ATH-IMI-CD-T-GM-DXT-AP	1		-	1	-	1	0.83
30	LEV-CRO-C-CIP-E-ATH-CD-T-GM-DXT-AP	11	5	5	10	1	16	0.92
31	CRO-C-CIP-E-ATH-IMI-CD-T-GM-DXT-AP	1		-	1	-	1	0.92
32	LEV-CRO-C-CIP-E-ATH-IMI-CD-T-GM-DXT-AP	7		3	3	1	7	1

**Table 4 antibiotics-09-00426-t004:** Multiple antibiotic resistance genes in *C*. *fetus*, *C. jejuni* and *C*. *coli* isolates.

No	Sample Source		*Campylobacter* Species			Multiple Resistance Genes Harbored
	Water Sample	Milk Sample	*C*. *jejuni*	*C*. *coli*	*C*. *fetus*	
1	+	-	2	-	-	*tetA*, *catII*
2	-	+	1	-	-	*catII*, *ermB*
3	-	-	1	-	1	*tetA*, *ampC*
4	-	+	-	-	1	*tetA*, *tetM*, *ampC*
5	+	-	-	-	1	*tetA*, *ampC*, *gyrA*
6	+	-	1	-	-	*tetK*, *ampC*, *catII*
7	+	-	1	-	-	*tetA*, *catII*, *gyrA*
8	+	-	6	-	-	*tetA*, *tetB*, *ampC*
9	+	+	4	-	-	*tetA*, *ampC*, *catII*
10	+	-	-	1	-	*tetM*, *ampC*, *gyrA*
11	-	+	-	2	-	*tetA*, *ampC*, *aac(3)-IIa-(aacC2)^a^*
12	-	+	-	-	1	*tetA*, *catII*, *aac(3)-IIa-(aacC2)^a^*
13	+	+	-	-	2	*tetA*, *tetM, ampC*, *catII*
14	+	-	2	-	-	*tetA*, *tetB*, *ampC*, *aac(3)-IIa-(aacC2)^a^*
15	+	-	1	-	-	*tetA*, *tetB, ampC*, *ermB*
16	+	-	1	-	-	*tetA*, *tetB, ampC*, *catII*
17	+	-	-	1	-	*tetM, ampC*, *catII, aac(3)-IIa-(aacC2)^a^*
18	+	-	1	-	-	*tetA*, *catII, ermB*, *aac(3)-IIa-(aacC2)^a^*
19	+	-	-	1	1	*tetA*, *ampC*, *gyrA*, *aac(3)-IIa-(aacC2)^a^*
20	+	-	2	-	-	*tetA*, *tetB*, *ampC*, *catII, gyrA*
21	+		4	3	-	*tetA*, *ampC*, *catII, gyrA*, *aac(3)-IIa-(aacC2)^a^*
22	+	-	1	-	-	*tetA*, *tetM, catII*, *ermB*, *gyrA*
23	+	-	1	-	-	*tetA*, *tetB*, *ampC*, *catII, gyrA*
24	-	+	-	1	-	*tetM*, *ampC*, *catII, gyrA*, *aac(3)-IIa-(aacC2)^a^*
25	+	+	2		2	*tetA*, *tetM*, *ampC*, *catII, aac(3)-IIa-(aacC2)^a^*
26	+	-	1	1	1	*tetA*, *tetM*, *ampC*, *catII, gyrA*
27	+	-	3	-	-	*tetA*, *tetB*, *ampC*, *ermB*, *aac(3)-IIa-(aacC2)^a^*
28	+	-	2	-	-	*tetA*, *tetB, ampC*, *ermB*, *gyrA*
29	+	-	-	1	-	*tetA*, *tetM, ampC*, *catII*, *ermB*, *aac(3)-IIa-(aacC2)^a^*
30	+	-	2	-	-	*tetA*, *tetM, ampC*, *catII*, *gyrA*, *aac(3)-IIa-(aacC2)^a^*

Note: + = presence, − = absence.

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
