# Peer review of "Occurrence, Virulence and Antimicrobial Resistance-Associated Markers in Campylobacter Species Isolated from Retail Fresh Milk and Water Samples in Two District Municipalities in the Eastern Cape Province, South Africa"

_antibiotics, 2020, doi:10.3390/antibiotics9070426_

Round 1

Reviewer 1 Report

Abstract is too lengthy, authors need to focus on key results only rather than presenting the data in the abstract.
In figure 1 authors showed only 8 positive isolates while in the study they got much positive. Please justify and explain it in the legend only which isolates these are and why only authors showed only these.
In figure 2 what are the numbers represent? Is it a percentage? Authors can make a better pie diagram for all the data they summarize in the abstract. Please make a pie diagram using the whole data and statistics. Authors can also include table 1 in that figure.
Authors should put all the gel images in one figure, making panels a, b, c……
What about the statistics in figure 8? How many times did authors perform this assay? Error bars are needed.
Please make the table number correct. Supplementary should be S1, S2….

Author Response

Response to Reviewers
Reviewer 1
Comment 1: Abstract is too lengthy; authors need to focus on key results only rather than presenting the data in the abstract
Response: The abstract has been summarized as directed by the reviewer.
Comment 2: In figure 1 authors showed only 8 positive isolates while in the study they got much positive. Please justify and explain it in the legend only which isolates these are and why only authors showed only these.
Response: Figure 1 is a representative gel image of some of the PCR confirmed genus Campylobacter.
Comment 3: In figure 2 what are the numbers represent? Is it a percentage? Authors can make a better pie diagram for all the data they summarize in the abstract. Please make a pie diagram using the whole data and statistics. Authors can also include table 1 in that figure.
Response: Figure 2 showed the pictorial representation of the distribution patterns of the number of presumptive and confirmed Campylobacter isolates obtained from different milk sources. S1 formal Table 1 cannot be represented as a pie chat because it showed the primer sets used study not results.
Comment 4: Authors should put all the gel images in one figure, making panels a, b, c…… What about the statistics in figure 8? How many times did authors perform this assay? Error bars are needed. Please make the table number correct. Supplementary should be S1, S2….
Response: All the gel pictures cannot be in one figure because some of the base pair are very close and it will be confusing and heard for readers to differentiate. Figure 8 is the result of antibiotic susceptibility pattern of isolates against the test drugs and it does not require error bars. Also, supplementary tables have been changed to S1 and S2 as seen in line 140 and 165-166 and the table numbers have been rearranged to suit the table numbers.

Reviewer 2 Report

The manuscript “Occurrence, virulence and antimicrobial resistance-associated markers in Campylobacter species isolated from retail fresh milk and water samples in two District Municipalities, in the Eastern Cape Province, South Africa” present an analysis of both antibiotic resistance, and the presence of ARGs and virulence factors from hundreds of presumptive Campylobacter isolates isolated from retail milk and water. Further identification confirms 162 isolates and only 70 of these further analyzed. Although the sampling number was large, the actual analyzed number (70) makes interpretation challenging.

Major changes/comments:

The authors acquired 128 samples from a variety of sources. This is an impressive number of sources; however, a major question is how many of those samples identified Campylobacter spp. as a pollutant? There should be a figure discussing “prevalence” of Campylobacter in various sources. For example, in Fig. 2 it appears milk samples from car/roads is a risk factor for campylobacter contamination but without showing prevalence (%) is difficult to conclude.

In line 186 it identifies 162 confirmed isolates without identifying the sources. Without these details, minimal conclusions can be drawn. It would be hypothesized that major differences exist among antibiotic resistance patterns from your water sampling sites and/or milk samplings, but without any further identification or categorization results remain inconclusive. 

In Table 3, there is a large % of campylobacter unaccounted for. The authors state 162 isolates (line 186) but only 70 are listed in Table 3, where are the other 92 and how did you pick the 70?

In section 2.5 regarding the isolation from milk samples, was the same volumes used for all milk samples? If so, please include volume tested. What was the contamination concentration of Campylobacter contamination (CFU/ml)?

In Table 4, it states prevalence in the title though it is unknown what the numbers in the tables are referring to. More details are needed and should include actual n# and %. Additionally, the number of isolates in each species analyzed should also be included.

In Table 4, the purpose of identify virulence factors is to help in assessing the potential to be pathogenic. The more virulence factors per isolate, potentially a greater threat. Were there any isolates exhibiting multiple virulence factors? What was the prevalence so that milk isolates could be compared to water isolates? Were there also isolates that demonstrated antibiotic resistance?

In Fig 8, there are no n# associated with any of the species. Also, we see “other Campylobacter” for the first time with no discussion as to what defines this category. Are they presumptive or confirmed? Line 230 states only 70 were analyzed for resistance; however, more information is needed of those 70. I am guessing it is the same 70 from Table 3, but then they shouldn’t be included in the same chart considering they are from different sources: milk vs water. Would suggest breaking these up and analyzing separately. Additionally, where are the isolates from water and milk acquired?  This will help in interpreting your findings.

The statement “least resistance” from line 262 does not agree with the data from line 256. Please explain.

The high prevalence of ceftriaxone is a serious public health concern. Were these isolates from water or milk? You analyzed Oxa-48, but did you look at common ESBLs like CTX-M or TEMs?

Figure 9 suggests demonstrates your positive control for aac(3) did not work. Please explain.

Line 292 state water was more contaminated, but in the previous sentence it states the same levels: 36.92 vs 37.11%. Please explain

Overall the discussion needs more details when referring to other published studies so that the reader understands the correlation.

In line 314, explain how your findings “exhibit notable differences in the spread of iam”.

Line 294 is not true the way the data is currently presented considering it is talking about %

In the discussion section, there should also be more discussion as to the relevance and importance of the virulence factors analyzed.

Minor changes:

Line 68: should read “disease outbreaks”

Line 70-72: The two sentences state the same thing. Delete one.

Line 82: “Antibiotic”, omit plural

Line 87: “environments are”

Line 89: “markers”

Line 116: please define what antibiotics comprise the “antibiotic selective supplement”

Line 136: “pairs”

Line 139: define what “modification genes” are

Line 141: where did your master mix come from?

Section 2.8: define zones of inhibition used for resistance and susceptibility

Line 206: replace modification with “regulator”

Line 235: space after “against”

Lines 290-292: clarify what are isolates vs samples in this sentence

Line 327-328: “quotations” are not used in research articles. Rather restate what the authors found in your own words and cite it

Line 336: harbor

Line 338-341: redundant lines

There is minimal need for the gels in the body of the manuscript.

Author Response

Reviewer 2
Comment: The authors acquired 128 samples from a variety of sources. This is an impressive number of sources; however, a major question is how many of those samples identified Campylobacter spp. as a pollutant? There should be a figure discussing “prevalence” of Campylobacter in various sources. For example, in Fig. 2 it appears milk samples from car/roads is a risk factor for campylobacter contamination but without showing prevalence (%) is difficult to conclude
Response: The result discussing “the prevalence” of Campylobacter in various sources are shown in section 3.1 in line 170-172. Figure 2 is pictorial representation of presumptive/confirmed Campylobacter isolates recovered from milk samples from different sources.
Comment 2: In line 186 it identifies 162 confirmed isolates without identifying the sources. Without these details, minimal conclusions can be drawn. It would be hypothesized that major differences exist among antibiotic resistance patterns from your water sampling sites and/or milk samplings, but without any further identification or categorization results remain inconclusive.
Response: The source of 162 identified genus Campylobacter isolates were clearly stated in line 168-171.
Comment 3: In Table 3, there is a large % of campylobacter unaccounted for. The authors state 162 isolates (line 186) but only 70 are listed in Table 3, where are the other 92 and how did you pick the 70?
Response: The 162 isolates mentioned in line 170 were identified as belonging to the genus Campylobacter. This genus Campylobacter were delineated into species and 12 isolates were identified to be C. fetus, 44 as C. jejuni and 14 as C. coli in total 70 as seen in Table 1 formal Table 3.
Comment 4: In section 2.5 regarding the isolation from milk samples, was the same volumes used for all milk samples? If so, please include volume tested. What was the contamination concentration of Campylobacter contamination (CFU/ml)?
Response: In section 2.5, the same volumes were used for all the milk samples and the volume used has been added as seen in line 117. Number of colonies on plates were not counted and was not among the objectives of the study.
Comment 5: In Table 4, it states prevalence in the title though it is unknown what the numbers in the tables are referring to. More details are needed and should include actual n# and %. Additionally, the number of isolates in each species analyzed should also be included.
Response: In Table 2, formal Table 4, the numbers in Table are the number of the various virulence genes detected and the percentages occurrence have been added. The number of each species detected were stated in Table 1.
Comment 6: In Table 4, the purpose of identify virulence factors is to help in assessing the potential to be pathogenic. The more virulence factors per isolate, potentially a greater threat. Were there any isolates exhibiting multiple virulence factors? What was the prevalence so that milk isolates could be compared to water isolates? Were there also isolates that demonstrated antibiotic resistance?
Response: In Table 2 formal Table 4, only showed the virulence genes detected in each species. Also, there were some isolates that exhibited multiple virulence factor as earlier explain in line 213-217 and also, we were not comparing the milk isolated against the water samples. Some of the isolates demonstrated antibiotics resistance as shown in Table 4.
Comment 7: In Fig 8, there are no n# associated with any of the species. Also, we see “other Campylobacter” for the first time with no discussion as to what defines this category. Are they presumptive or confirmed? Line 230 states only 70 were analyzed for resistance; however, more information is needed of those 70. I am guessing it is the same 70 from Table 3, but then they shouldn’t be included in the same chart considering they are from different sources: milk vs water. Would suggest breaking these up and analyzing separately. Additionally, where are the isolates from water and milk acquired? This will help in interpreting your findings.
Response: Figure 8 is the result of antibiotic susceptibility pattern of the isolates against the 162 isolates. The 162 Campylobacter isolates mentioned in line 231 formal line 230 were the number of confirmed genus Campylobacter obtained from water and milk samples. The 70 isolates were among the 162 isolates and were the only isolates focused on because 92 isolates were not identified as belonging to any target species and the 92 isolates were classified as “other Campylobacter species” as shown in Figure 8. Also, we are not comparing water isolates against milk isolates.
Comment 8: The statement “least resistance” from line 262 does not agree with the data from line 256. Please explain.
Response: The statement is not referring to line 263 formal line 262 but referring to Table 4 which showed numbers of multiple resistance genes harbored in C. coli and C. jejuni. However, the word “multiple” has been added to the sentence in line 263 to give a better understanding of the statement.
Comment 9: The high prevalence of ceftriaxone is a serious public health concern. Were these isolates from water or milk? You analyzed Oxa-48, but did you look at common ESBLs like CTX-M or TEMs?
Response: The isolates from both water and milk samples were resistance to ceftriaxone. Also, the isolates that were resistance to imipenem were screened for only Oxa-48 gene.
Comment 10: Figure 9 suggests demonstrates your positive control for aac(3) did not work. Please explain.
Response: Figure 9 is a representative electrophoresis picture of various amplified antibiotics resistance genes detected in Campylobacter species. Lane 6 showed the band size of aac(3)-IIa (aacC2)a gene not positive control.
Comment 11: Line 292 state water was more contaminated, but in the previous sentence it states the same levels: 36.92 vs 37.11%. Please explain?
Response: In line 293-295, the sentence has been changed to give a better understanding to the reader but the figure is same.
Comment 12: Overall the discussion needs more details when referring to other published studies so that the reader understands the correlation.
Response: The discussion section has enough information and references in supporting our finding. Also, the information in the discussion section is enough to give the reader a better understanding of the study.
Comment 13: In line 314, explain how your findings “exhibit notable differences in the spread of iam”.
Response: The sentence has been rephrase as seen in line 316 formal line 314.
Comment 14: Line 294 is not true the way the data is currently presented considering it is talking about % Response: The statement has been changed as seen in line 293-295.
Comment 15: In the discussion section, there should also be more discussion as to the relevance and importance of the virulence factors analyzed.
Response: The detailed information about the importance of the virulence genes screened have been reported in line 317, 321-322, 329, 331-332.
Minor changes
Comment 16: Line 68: should read “disease outbreaks”
Response: In line 69 formal line 68 “disease outbreaks” has been added as directed by the reviewer.
Comment 17: Line 70-72: The two sentences state the same thing. Delete one.
Response: In line 70-72, the two statement are not saying the same thing. One is referring to the common symptoms while the other is talking about extreme case of infection.
Comment 18: Line 82: “Antibiotic”, omit plural
Response: In line 78 formal line 82, “antibiotics” has been changed to antibiotic as directed by the reviewer.
Comment 19: Line 87: “environments are”
Response: In line 83, formal line 87 the sentence “environment is” has been changed to “environments are” as directed by the reviewer.
Comment 20: Line 89: “markers”
Response: In line 85 formal line 89, the word “markers” is now correctly spelled.
Comment 21: Line 116: please define what antibiotics comprise the “antibiotic selective supplement”
Response: The composition of the antibiotic selective supplement has been added in line 112 formal line 116.
Comment 22: Line 136: “pairs”
Response: In line 133 formal line 136, the word pairs is now correctly spelled.
Comment 23: Line 139: define what “modification genes” are. Response: It is not necessary to definition every word used and I don’t think the definition is necessary in this contest.
Comment 24: Line 141: where did your master mix come from?
Response: The place where the master mix was bought has been added as seen in line 138 formal line 141.
Comment 24: Section 2.8: define zones of inhibition used for resistance and susceptibility.
Response: The CLSI 2015 guild line was used for the interpretation of antibiotic resistance result and it is not necessary to define the zone of inhibition and susceptibility result of each antibiotics used.
Comment 25: Line 206: replace modification with “regulator”
Response: In Line 206: the word “modification” has been replaced with “regulator as directed by the reviewer.
Comment 26: Line 235: space after “against
Response: In Line 235: there is space after “against”
Comment 27: Lines 290-292: clarify what are isolates vs samples in this sentence
Response: In line 293-194, the sentence has been modified to give a clear definition between isolates and samples.
Comment 28: Line 327-328: “quotations” are not used in research articles. Rather restate what the authors found in your own words and cite it.
Response: In line 329-330, the quotation signs have been removed and the sentence has been restated in my own words.
Comment 29: Line 336: harbor
Response: In line 338 formal line 336, the word “harbored” has been changed to harbor as directed by the reviewer.
Comment 30: Line 338-341: redundant line
Response: Line 338-339 is not redundant because it is supporting our finding that Campylobacter species harbor multiple virulence gene.

Round 2

Reviewer 1 Report

How many times did authors perform the assays?

Authors should add the statistics section in material and methods and should apply to the whole manuscript.

Author Response

Response to Reviewers

Comment 1: How many times did authors perform this assay?

Response: Antibiotic susceptibility assay was carried out once. This assay is to determine the potency of the antibiotic used and does not required to be carried out more than one because is a standard antibiotic not plant extract that is required to be carried out in triplicates.

Comment 2: Authors should add the statistics section in material and methods and should apply to the whole manuscript.

Response: In line 169-170, statistical analysis has been included in material and method as directed by the reviewer.

Reviewer 2 Report

Reviewer 2

Comment: The authors acquired 128 samples from a variety of sources. This is an impressive number of sources; however, a major question is how many of those samples identified Campylobacter spp. as a pollutant? There should be a figure discussing “prevalence” of Campylobacter in various sources. For example, in Fig. 2 it appears milk samples from car/roads is a risk factor for campylobacter contamination but without showing prevalence (%) is difficult to conclude

Response: The result discussing “the prevalence” of Campylobacter in various sources are shown in section 3.1 in line 170-172. Figure 2 is pictorial representation of presumptive/confirmed Campylobacter isolates recovered from milk samples from different sources.

Reviewer’s ResponseThe authors still haven’t addressed the comment of “prevalence”, thus meaning prevalence of Campylobacter in each sample. Lines 170-172 discuss their origin NOT the prevalence. The question is important considering all of your samples might have come from one sample. For example, you mention 72 different milk samplings, but only show 159 isolates, which is roughly 2 samples/site. How many of the milk samples were positive for Campylobacter? Additionally, 56 water samples with 279 isolates, was each sample always positive for Campylobacter? From a epidemiology perspective, this would be of interest to the reader

Comment 2: In line 186 it identifies 162 confirmed isolates without identifying the sources. Without these details, minimal conclusions can be drawn. It would be hypothesized that major differences exist among antibiotic resistance patterns from your water sampling sites and/or milk samplings, but without any further identification or categorization results remain inconclusive.

Response: The source of 162 identified genus Campylobacter isolates were clearly stated in line 168-171.

Reviewer’s Response. Same previous statement again. Also, include a n# in Table 3 and “other C. spp” to include the strains that were not identified.

Comment 3: In Table 3, there is a large % of campylobacter unaccounted for. The authors state 162 isolates (line 186) but only 70 are listed in Table 3, where are the other 92 and how did you pick the 70?

Response: The 162 isolates mentioned in line 170 were identified as belonging to the genus Campylobacter. This genus Campylobacter were delineated into species and 12 isolates were identified to be C. fetus, 44 as C. jejuni and 14 as C. coli in total 70 as seen in Table 1 formal Table 3.

See comment above (#2) to address this well

Comment 4: In section 2.5 regarding the isolation from milk samples, was the same volumes used for all milk samples? If so, please include volume tested. What was the contamination concentration of Campylobacter contamination (CFU/ml)?

Response: In section 2.5, the same volumes were used for all the milk samples and the volume used has been added as seen in line 117. Number of colonies on plates were not counted and was not among the objectives of the study.

There still is not a volume. Please  provide the amount of ml

Comment 5: In Table 4, it states prevalence in the title though it is unknown what the numbers in the tables are referring to. More details are needed and should include actual n# and %. Additionally, the number of isolates in each species analyzed should also be included.

Response: In Table 2, formal Table 4, the numbers in Table are the number of the various virulence genes detected and the percentages occurrence have been added. The number of each species detected were stated in Table 1.

Ok, some discussion or inclusion in the results/discussion should talk about the large variability between flgR between water and milk samples.

Comment 6: In Table 4, the purpose of identify virulence factors is to help in assessing the potential to be pathogenic. The more virulence factors per isolate, potentially a greater threat. Were there any isolates exhibiting multiple virulence factors? What was the prevalence so that milk isolates could be compared to water isolates? Were there also isolates that demonstrated antibiotic resistance?

Response: In Table 2 formal Table 4, only showed the virulence genes detected in each species. Also, there were some isolates that exhibited multiple virulence factor as earlier explain in line 213-217 and also, we were not comparing the milk isolated against the water samples. Some of the isolates demonstrated antibiotics resistance as shown in Table 4.

Comment 7: In Fig 8, there are no n# associated with any of the species. Also, we see “other Campylobacter” for the first time with no discussion as to what defines this category. Are they presumptive or confirmed? Line 230 states only 70 were analyzed for resistance; however, more information is needed of those 70. I am guessing it is the same 70 from Table 3, but then they shouldn’t be included in the same chart considering they are from different sources: milk vs water. Would suggest breaking these up and analyzing separately. Additionally, where are the isolates from water and milk acquired? This will help in interpreting your findings.

Response: Figure 8 is the result of antibiotic susceptibility pattern of the isolates against the 162 isolates. The 162 Campylobacter isolates mentioned in line 231 formal line 230 were the number of confirmed genus Campylobacter obtained from water and milk samples. The 70 isolates were among the 162 isolates and were the only isolates focused on because 92 isolates were not identified as belonging to any target species and the 92 isolates were classified as “other Campylobacter species” as shown in Figure 8. Also, we are not comparing water isolates against milk isolates.

Considering the environmental and animal sources, it would definitely benefit the reader to see which is more resistant or if significantly different

Comment 8: The statement “least resistance” from line 262 does not agree with the data from line 256. Please explain.

Response: The statement is not referring to line 263 formal line 262 but referring to Table 4 which showed numbers of multiple resistance genes harbored in C. coli and C. jejuni. However, the word “multiple” has been added to the sentence in line 263 to give a better understanding of the statement.

This is still unclear to what the authors’ are trying to state here. Are you talking about prevalence? If so the statement doesn’t agree with the data. Please clarify .

Comment 9: The high prevalence of ceftriaxone is a serious public health concern. Were these isolates from water or milk? You analyzed Oxa-48, but did you look at common ESBLs like CTX-M or TEMs?

Response: The isolates from both water and milk samples were resistance to ceftriaxone. Also, the isolates that were resistance to imipenem were screened for only Oxa-48 gene.

Analyzing water vs milk would help in understanding the public health threat; otherwise these findings are difficult to assess the impact on health

Comment 10: Figure 9 suggests demonstrates your positive control for aac(3) did not work. Please explain.

Response: Figure 9 is a representative electrophoresis picture of various amplified antibiotics resistance genes detected in Campylobacter species. Lane 6 showed the band size of aac(3)- IIa (aacC2)a gene not positive control.

Comment 11: Line 292 state water was more contaminated, but in the previous sentence it states the same levels: 36.92 vs 37.11%. Please explain?

Response: In line 293-295, the sentence has been changed to give a better understanding to the reader but the figure is same.

The sentence is confusing. It is also in the results section but the numbers 103/279 =36.92% does not state the prevalence of Campylobacter water or milk but rather the percentage of confirmed/presumptive in the corresponding sampling source. As stated above, please inform the reader how many of the milk and water samples were positive for campylobacter. This would tell us the %

Comment 12: Overall the discussion needs more details when referring to other published studies so that the reader understands the correlation.

Response: The discussion section has enough information and references in supporting our finding. Also, the information in the discussion section is enough to give the reader a better understanding of the study.

Comment 13: In line 314, explain how your findings “exhibit notable differences in the spread of iam”.

Response: The sentence has been rephrase as seen in line 316 formal line 314.

Comment 14: Line 294 is not true the way the data is currently presented considering it is talking about %

Response: The statement has been changed as seen in line 293-295.

Lines 294-296 state the milk was more contaminated, but  there is no data supporting it. The % given on Line 293 are not relevant to the statement being made in 294-296. Additionally, on line 298 you state your findings agree with others regarding high detection rates, but your manuscript does not have how many of your water samples were contaminated

Comment 15: In the discussion section, there should also be more discussion as to the relevance and importance of the virulence factors analyzed.

Response: The detailed information about the importance of the virulence genes screened have been reported in line 317, 321-322, 329, 331-332.

Minor changes

Comment 16: Line 68: should read “disease outbreaks”

Response: In line 69 formal line 68 “disease outbreaks” has been added as directed by the reviewer.

Comment 17: Line 70-72: The two sentences state the same thing. Delete one.

Response: In line 70-72, the two statement are not saying the same thing. One is referring to the common symptoms while the other is talking about extreme case of infection.

Comment 18: Line 82: “Antibiotic”, omit plural

Response: In line 78 formal line 82, “antibiotics” has been changed to antibiotic as directed by the reviewer.

Comment 19: Line 87: “environments are”

Response: In line 83, formal line 87 the sentence “environment is” has been changed to “environments are” as directed by the reviewer.

Comment 20: Line 89: “markers”

Response: In line 85 formal line 89, the word “markers” is now correctly spelled.

Comment 21: Line 116: please define what antibiotics comprise the “antibiotic selective supplement”

Response: The composition of the antibiotic selective supplement has been added in line 112 formal line 116.

Comment 22: Line 136: “pairs”

Response: In line 133 formal line 136, the word pairs is now correctly spelled.

Comment 23: Line 139: define what “modification genes” are.

Response: It is not necessary to definition every word used and I don’t think the definition is necessary in this contest.

Comment 24: Line 141: where did your master mix come from?

Response: The place where the master mix was bought has been added as seen in line 138 formal line 141.

Comment 24: Section 2.8: define zones of inhibition used for resistance and susceptibility. Response: The CLSI 2015 guild line was used for the interpretation of antibiotic resistance result and it is not necessary to define the zone of inhibition and susceptibility result of each antibiotics used.

Comment 25: Line 206: replace modification with “regulator”

Response: In Line 206: the word “modification” has been replaced with “regulator as directed by the reviewer.

Comment 26: Line 235: space after “against

Response: In Line 235: there is space after “against”

Comment 27: Lines 290-292: clarify what are isolates vs samples in this sentence

Response: In line 293-194, the sentence has been modified to give a clear definition between isolates and samples.

Comment 28: Line 327-328: “quotations” are not used in research articles. Rather restate what the authors found in your own words and cite it.

Response: In line 329-330, the quotation signs have been removed and the sentence has been restated in my own words.

Comment 29: Line 336: harbor

Response: In line 338 formal line 336, the word “harbored” has been changed to harbor as directed by the reviewer.

Comment 30: Line 338-341: redundant line

Response: Line 338-339 is not redundant because it is supporting our finding that

Campylobacter species harbor multiple virulence gene.

Author Response

Reviewer 2

Comment 1: The authors still haven’t addressed the comment of “prevalence”, thus meaning prevalence of Campylobacter in each sample. Lines 170-172 discuss their origin NOT the prevalence. The question is important considering all of your samples might have come from one sample. For example, you mention 72 different milk samplings, but only show 159 isolates, which is roughly 2 samples/site. How many of the milk samples were positive for Campylobacter? Additionally, 56 water samples with 279 isolates, was each sample always positive for Campylobacter? From a epidemiology perspective, this would be of interest to the reader.

Response: The result discussing the origin of the samples “the prevalence” of Campylobacter in various sources and the number of milk samples positive for Campylobacter are shown in line 174-176. However, the number of water and milk samples that was positive for Campylobacter is as seen in line 299-302.

Comment 2: Same previous statement again. Also, include a n# in Table 3 and “other C. spp” to include the strains that were not identified.

Response: The source of the 70 isolates focused on has been added as seen in Table 3. Also, the pattern of resistance to other species can not be included in this table because our focus is not on them and it is not necessary.

Comment 3: In Table 3, there is a large % of campylobacter unaccounted for. The authors state 162 isolates (line 186) but only 70 are listed in Table 3, where are the other 92 and how did you pick the 70?

Response: The 162 isolates mentioned in line 191 were delineated into species and the 70 isolates selected belong to C. fetus, C. jejuni and C. coli as seen in Table 1 formal Table 3 which are the main target species in this study implicated in human infection. There are so many other species such as C. hyointestinalis, C. upsaliensis, C. ureolyticus, C. concisus, C. helveticus, C. insulaenigrae, C. peloridis, C. hominis, C. gracilis, C. lanienae, C. mucosalis, and C. sputorum which were not screened for and were classified as other Campylobacter species in this study. The 92 isolates have been included in table 1 formal table 3 as requested by the reviewer.

Comment 4: In section 2.5 regarding the isolation from milk samples, please provide the amount of ml.

Response: The volume of the milk samples used has been stated in line 117.

Comment 5: Ok, some discussion or inclusion in the results/discussion should talk about the large variability between flgR between water and milk samples.

Response: In line 334-336, has explain variability of detection of flgR gene between water and milk samples.

Comment 6: In Fig 8, Considering the environmental and animal sources, it would definitely benefit the reader to see which is more resistant or if significantly different.

Response: In Figure 8, we are not comparing water isolates against milk isolates and Table 3 has shown which source is more resistant to the antibiotics used.

Comment 7: The statement “least resistance” from line 262 does not agree with the data from line 256. This is still unclear to what the authors’ are trying to state here. Are you talking about prevalence? If so the statement doesn’t agree with the data. Please explain.

Response: The statement has been removed to prevent confusion.

Comment 8: The high prevalence of ceftriaxone is a serious public health concern. Were these isolates from water or milk? Analyzing water vs milk would help in understanding the public health threat; otherwise these findings are difficult to assess the impact on health.

Response: Table 3 shows the sources and the number of isolates resistance to ceftriaxone.

Comment 9: Line 292 state water was more contaminated, but in the previous sentence it states the same levels: 36.92 vs 37.11%. Please explain? The sentence is confusing. It is also in the results section but the numbers 103/279 =36.92% does not state the prevalence of Campylobacter water or milk but rather the percentage of confirmed/presumptive in the corresponding sampling source. As stated above, please inform the reader how many of the milk and water samples were positive for campylobacter. This would tell us the %

Response: In line 299-302, the number of milk and water samples positive for Campylobacter has been added.

Comment 10: Lines 294-296 state the milk was more contaminated, but there is no data supporting it. The % given on Line 293 are not relevant to the statement being made in 294-296. Additionally, on line 298 you state your findings agree with others regarding high detection rates, but your manuscript does not have how many of your water samples were contaminated.

Response: The number of water samples contaminated with Campylobacter has been added in line 299-302 as directed by the reviewer.

Round 3

Reviewer 2 Report

Reviewer 2 see below responses below

Comment 1: The authors still haven’t addressed the comment of “prevalence”, thus meaning prevalence of Campylobacter in each sample. Lines 170-172 discuss their origin NOT the prevalence. The question is important considering all of your samples might have come from one sample. For example, you mention 72 different milk samplings, but only show 159 isolates, which is roughly 2 samples/site. How many of the milk samples were positive for Campylobacter? Additionally, 56 water samples with 279 isolates, was each sample always positive for Campylobacter? From a epidemiology perspective, this would be of interest to the reader.

Response: The result discussing the origin of the samples “the prevalence” of Campylobacter in various sources and the number of milk samples positive for Campylobacter are shown in line 174-176. However, the number of water and milk samples that was positive for Campylobacter is as seen in line 299-302.

Your response does not address my comment; however, you have provided these analyses in lines 299-302. Please place in the results section

Comment 5: Ok, some discussion or inclusion in the results/discussion should talk about the large variability between flgR between water and milk samples.

Response: In line 334-336, has explain variability of detection of flgR gene between water and milk samples.

Table 2 needs to include n# otherwise it is difficult to understand where % is coming from. Look at Table 2 and look at flgR, it is 71% in milk compared to 7% in water. This does not agree with the text. Please address

Comment 6: In Fig 8, Considering the environmental and animal sources, it would definitely benefit the reader to see which is more resistant or if significantly different.

Response: In Figure 8, we are not comparing water isolates against milk isolates and Table 3 has shown which source is more resistant to the antibiotics used.

Then what is the conclusion? Comparing milk to environmental samples seems like the purpose of your study and yet there is no comparison.

Comment 8: The high prevalence of ceftriaxone is a serious public health concern. Were these isolates from water or milk? Analyzing water vs milk would help in understanding the public health threat; otherwise these findings are difficult to assess the impact on health.

Response: Table 3 shows the sources and the number of isolates resistance to ceftriaxone.

Reporting the prevalence of resistance is the key points, otherwise how can it be compared to other studies.

Author Response

Response to Reviewers 2

Comments 1: Your response does not address my comment; however, you have provided these analyses in lines 299-302. Please place in the results section.

Response: In line 175-179, the analyses has been placed in the result section as directed by the reviewer.

Comment 2: Table 2 needs to include n# otherwise it is difficult to understand where % is coming from. Look at Table 2 and look at flgR, it is 71% in milk compared to 7% in water. This does not agree with the text. Please address.

Comments: The number of isolates has been added as directed by the reviewer which will help in understanding the percentage values. In line 334-335, percentage values of flgR gene detection has been address.

Comment 3: Then what is the conclusion? Comparing milk to environmental samples seems like the purpose of your study and yet there is no comparison.

Responses: Table 3 shows the result of the differences in the pattern of antimicrobial phenotypic resistance profile observed in isolates obtained from environmental and animal sources.

Comment 4: Reporting the prevalence of resistance is the key points, otherwise how can it be compared to other studies.

Responses: In line 369-376, the prevalence of antibiotic resistance results obtained in this study were compared with other studies.
